# Intermolecular Interactions in 3-Aminopropyltrimethoxysilane, N-Methyl-3-aminopropyltrimethoxysilane and 3-Aminopropyltriethoxysilane: Insights from Computational Spectroscopy

**DOI:** 10.3390/ijms242316634

**Published:** 2023-11-23

**Authors:** Mariela M. Nolasco, Stewart F. Parker, Pedro D. Vaz, Paulo J. A. Ribeiro-Claro

**Affiliations:** 1CICECO—Instituto de Materiais de Aveiro, Departamento de Química, Universidade de Aveiro, 3810-193 Aveiro, Portugal; prc@ua.pt; 2ISIS Neutron & Muon Source, STFC Rutherford Appleton Laboratory, Didcot OX11 0QX, UK; stewart.parker@stfc.ac.uk; 3Champalimaud Foundation, Champalimaud Centre for the Unknown, 1400-038 Lisboa, Portugal; pedro.vaz@fundacaochampalimaud.pt

**Keywords:** non-covalent interactions, hydrogen bonds, DFT calculations, cluster model, inelastic neutron scattering (INS)

## Abstract

In this work, a computational spectroscopy approach was used to provide a complete assignment of the inelastic neutron scattering spectra of three title alkoxysilane derivatives—3-aminopropyltrimethoxysilane (APTS), N-methyl-3-aminopropyltrimethoxysilane (MAPTS), and 3-aminopropyltriethoxysilane (APTES). The simulated spectra obtained from density functional theory (DFT) calculations exhibit a remarkable match with the experimental spectra. The description of the experimental band profiles improves as the number of molecules considered in the theoretical model increases, from monomers to trimers. This highlights the significance of incorporating non-covalent interactions, encompassing classical NH···N, N–H···O, as well as C–H···N and C–H···O hydrogen bond contacts, to achieve a comprehensive understanding of the system. A distinct scenario emerges when considering optical vibrational techniques, infrared and Raman spectroscopy. In these instances, the monomer model provides a reasonable description of the experimental spectra, and no substantial alterations are observed in the simulated spectra when employing dimer and trimer models. This observation underscores the distinctive ability of neutron spectroscopy in combination with DFT calculations in assessing the structure and dynamics of molecular materials.

## 1. Introduction

(3-Aminopropyl)trimethoxysilane, hereafter referred to as APTS, (methyl-3-aminopropyl)trimethoxysilane (MAPTS) and (3-aminopropyl)triethoxysilane (APTES), Figure 1, are organosiloxane compounds that play a significant role in various industries and applications (see, e.g., [1,2,3,4,5,6,7]). 

Their importance stems from their use in surface modification via creating a thin layer of amino-functionalized silane. This layer can improve surface wetting, reduce surface tension, and enhance the compatibility between materials. It finds applications in industries like textiles, paints, and plastics, where improved surface properties are desired. The same applies in the biomedical and pharmaceutical fields, as the surface of biomaterials, such as implants or drug delivery systems, are used to improve biocompatibility, reduce immunogenicity, and enhance the interaction with biological tissues (see, e.g., [7,8,9,10,11,12,13,14,15,16,17,18,19,20,21]).

Although somewhat similar in their backbone structures, all three compounds have found distinct applications. For instance, APTES was used to modify the surface of ZnO quantum dots and enhance its fluorescence emission, while improving water dispersion properties [11]. Its application as a thermal insulator has also been studied. In this context, APTES was used for improving the fire resistance of surface-coated *Araucaria angustifolia* (Brazilian pine tree) wood [12] and to enhance the heat resistance properties of epoxy composites to high temperatures [13]. In this field, more recently, it has been used to prepare co-polymers to coordinate with Eu^3+^ emissive complexes without affecting lanthanide’s emission (615 nm), making it stable up to 300 °C [14]. In a study, APTES and MAPTS were used to derivatize the surface of mesoporous alumina towards its use in the removal of iodine under a radioactive setting for safe geological disposal of radioactive nuclei [15]. MAPTS has also successfully been used in the field of health applications. A study by Lee et al. used it as a surface modifier of fibers for improving NO release properties in hypoxia/reoxygenation injury treatment [16]. In another study, it was used as a modifier in dental fillers to minimize the shrinkage stress experienced by unmodified silica materials, thus improving the lifetime and performance of the filler materials [17]. In the industrial and environmental fields, MAPTS was reported as a modifier component of biogenic silica membranes for enhancing the separation properties of CO_2_/N_2_ mixtures [18]. In addition, APTS was used as a surface modifier of cellulose nanofibers to control their water dispersion and biodegradation without altering the structure or crystallinity of the nanofibers [19]. It was also used to prepare polyurethane co-polymers with the objective of promoting controlled release of fertilizers, while maintaining the integrity of the guest species against moisture and heat [20]. MAPTS was also used in the field of energy materials. Wang et al. used it to passivate the surface of CsPbBr_3_ perovskite nanocrystals using two alternative methods—post-synthesis or during hot-injection synthesis of the nanocrystals. Both methods yielded perovskite nanocrystals with enhanced stability, delivering stable and high-purity color conversion that could be tuned using the halide exchange capability of the nanocrystals that was preserved after passivation with APTS [21].

In recent years, inelastic neutron scattering (INS) spectroscopy has emerged as a characterization technique for materials, mainly due to notable improvements in spectrometer sensitivity, e.g., LAGRANGE at ILL [22], TOSCA at ISIS [23], and VISION at Oak Ridge [24], which has made this spectroscopic technique more amenable to small quantities and more structured samples. Examples of the application of this spectroscopic technique across a wide range of different systems can be found in the literature [25,26,27,28,29,30,31]. In all cases, the interpretation of the experimental INS spectra takes advantage of the ease of generating INS spectra from quantum mechanical frequency calculations. Owing to the absence of selection rules in INS spectroscopy, the linear dependence of intensities on atomic properties (scattering cross section), and the amplitude of vibrational motions, it is possible to break down the INS spectrum into distinct populations. Each of these populations can then be accurately modeled individually using density functional theory (DFT), leading to reliable and mostly unambiguous assignment of the INS spectrum.

The main objective of this study was to assign INS spectra of APTS, MAPTS, and APTES, with a specific emphasis on their potential application in analyzing surface derivatized materials. Additionally, this study aims to elucidate the significance of intermolecular, non-covalent interactions in providing a comprehensive understanding of these systems. Non-covalent interactions—namely strong and weak hydrogen bonds—are expected to play a significant role in the condensed phases of hydrocarbon compounds bearing nitrogen and oxygen atoms. The comparison between INS spectroscopy and its optical counterparts (IR and Raman spectroscopy) also opens a discussion about intensity calculations, which is a crucial aspect in computational spectroscopy.

## 2. Results and Discussion 

The computational spectroscopy approach applied in this work aimed at providing predictions of spectroscopic properties based on a reliable computational model. As is discussed in Section 2.2 below, the “single molecule” model fails to provide a satisfactory description of the experimental INS spectrum—which is not a surprise for a system comprising significant intermolecular, or non-covalent, interactions. Such description of the experimental features is obtained when considering a “cluster model” with up to three molecules, and the calculated structures for the dimer and trimer models are presented in Section 2.1 below. These clusters improve the description of the systems by considering the existence of intermolecular interactions, which are not to be ignored in this context.

It should be mentioned that larger clusters (tetramers, pentamers, …) were not considered given that trimers offer the best compromise between the calculated properties and the experimental data at a relatively low computational effort. Although it is always possible to extend the computational effort (higher clusters, larger basis sets, improved functionals), a limit must be set when a computational spectroscopy goal is achieved.

### 2.1. Calculated Structures: From Monomer to Trimer

Figure 1A,B, show the DFT-optimized structures for APTS and MAPTS monomers and two illustrative structures of dimers. For each system (APTS and MAPTS), two dimer structures were optimized: head-to-head (labeled HH) and head-to-tail (labeled HT). Figure 1C, presents the optimized structures of HT and HH trimers of APTES. Similar HT and HH structures were obtained for APTS and MAPTS. The energies of the optimized structures shown in Appendix A.

As evidenced in Figure 1 and Table 1, the presence of N–H···N hydrogen bonds is a key feature in stabilizing the head-to-head structures. The strength of N–H···N interactions increases from dimer to trimer (contact distances decrease from ca. 225 pm to below 220 pm), an effect that can be related to cooperativity, possible in the trimer forms. In the case of head-to-tail structures, the N–H···O hydrogen bonds are the most important intermolecular interactions. Other types of hydrogen bond contacts can be found in the optimized dimer and trimer structures, which further contribute to the stabilization of the clusters. For instance, C–H···O contacts are observed for all head-to-head structures. In the case of trimers, C–H···N interactions are also found to contribute to the stabilization of head-to-tail molecular associations. Of course, van der Waals interactions are expected to play a role in all clusters. 

### 2.2. Vibrational Spectroscopy

Figure 2 presents the IR, Raman, and INS spectra of the APTS sample, evidencing the complementarity of the information obtained from the three vibrational techniques, using photons and neutrons, respectively.

Due to the absence of symmetry-related selection rules mentioned above, INS has the advantage of being able to observe all vibrational modes of a system (although they can be either weak or fall in an overcrowded region, thus preventing their clear observation in the experimental spectrum). As can be seen in Figure 2, there are at least two modes of APTS, inactive or weak, in the optical techniques that are readily observable with neutron spectroscopy (components of the deformation modes CH_2_ wag and CH_3_ rock). Thus, the full assignment of the vibrational modes of any system must profit from this complementarity of spectral information. 

DFT calculations can provide a reliable description of the INS spectra of a molecular system. This is because DFT can be used to calculate both the frequency and intensity of each vibrational mode with high accuracy. In contrast, the intensities calculated in optical spectroscopy techniques (infrared and Raman) depend on a complex evaluation of how the electronic properties of the system change during vibrations (dipole moment and polarizability, respectively). On the other hand, for INS spectroscopy, the intensity of a band associated with a vibrational mode *k*, for a given atom *i*, is expressed by the dynamic structure factor [32,33].
(1)$$S_{i}^{*}\left(Q,n\nu_{k}\right)\propto \frac{{\left({QU}_{i}\right)}^{2n\;\;}}{n!}.\sigma_{i}.\;exp\left(-Q^{2}\alpha_{i}^{2}\right)$$
where *Q* is the momentum transferred to the sample, *n* is the quanta involved, ν*_k_* is the energy of the vibrational mode *k*, *U_i_* is the displacement vector of atom *i* in mode *k*, *σ_i_* is the neutron scattering cross section of atom *i*, and *α_i_* is a term related to the displacements of the atom in all vibrational modes. In this way, INS intensities are a function of two main factors: neutron scattering cross-section *σ_i_* and the atomic displacements in normal modes *U_i_*. While the neutron scattering cross-section is a physical property of the atom, the atomic displacements in normal modes are a straightforward result of the vibrational frequency calculations. So, it is possible to predict INS intensities from quantum chemical calculations with high accuracy, allowing for mostly unambiguous assignments of INS spectra, as already shown for diverse systems [25,26,27,28,29,30,31].

Figure 3 compares the experimental and calculated INS spectra of APTS in the 100–2000 cm^−1^ range (the experimental INS spectrum up to 4000 cm^−1^ is shown in Appendix A). It should be mentioned that discrete (or molecular) calculations cannot account for the low wavenumber crystal lattice or external vibrational modes. These modes fall usually below ca. 100 cm^−1^, but their influence can be felt at higher wavenumbers. However, it is shown that for liquids with appreciable viscosity, a “shock freezing” procedure (as described in the Materials and Methods below) can be used to create an amorphous phase that keeps the liquid phase morphology, which distinctly differs from the crystal structure. In such a case, periodic calculations are not feasible, and discrete calculations are the correct approach, as used herein. 

It can be concluded from Figure 3 that the experimental INS spectrum is poorly described by the simulated spectrum of the isolated monomer. Some of the main spectral features are present in the monomer spectrum, but the general intensity profile is far from satisfactory. Taking two illustrative examples, the intensity of the δCH_3_ modes at ca. 1500 cm^−1^ is clearly underestimated, while the intensity of the νSi-C mode at ca. 850 cm^−1^ is overestimated. 

The quality of the prediction increases when considering the head-to-head dimer, namely, in the region above 1000 cm^−1^, signaling the importance of intermolecular contacts. The best agreement is obtained with the head-to-tail trimer model. Even the low wavenumber region is surprisingly well predicted from the trimer model using broader bands in the simulation (calculated spectra segment labeled (a) in Figure 3). This is consistent with the amorphous nature of the sample. 

The best prediction of the experimental INS spectrum of MAPTS is also obtained from a trimer model. As shown in Figure 4, both trimer forms offer a reasonable simulated spectrum. The presence of the additional methyl group is observed in the intensity profile at ca. 980 cm^−1^ and 1090 cm^−1^ (N–CH_3_ rocking modes) and from the individual band at ca. 475 cm^−1^, related to the C–N–CH_3_ skeletal bending mode. The calculated spectra display a split of the CH_3_ torsional modes, assigned to “free” and “H-bonded” CH_3_ rotors, which is not clearly observed in the experimental spectrum (Figure 4).

The description of the INS spectrum of APTES follows the same trends (Figure 5). Important features are predicted by the calculated spectrum of the monomer (e.g., the CH_2_ wag + CH_3_ rock band at ca. 808 cm^−1^), but there are also important discrepancies (for instance, the predicted intense band at ca. 850 cm^−1^ is not observed experimentally). As a general trend, the description improves with increasing cluster size. A notable feature in the calculated spectra is the prediction of the strong broad band at ca. 267 cm^−1^, ascribed to the torsional motion of the methyl groups. Nonetheless, as can be seen from Figure 5, the prediction of the intensity contributions from CH_2_ and CH_3_ bending modes (namely, rock, twist, and wag modes) is not as good as for the previous systems. This seems to be related to the additional conformational freedom of the APTES skeleton (relative to APTS). Replacing methyl groups with ethyl groups generates multiple combinations of trans and gauche orientations around C–C and C–O single bonds. In the liquid phase at room temperature, a large number of these conformations are expected, but a full search of the conformational landscape of APTES was beyond the scope of this work.

The remarkable agreement between the observed INS spectrum and that estimated from the cluster of three molecules allows for a confident assignment of the vibrational modes of APTS, MAPTS, and APTES based on the selected clusters. To the best of our knowledge, no such assignment has been reported for these systems. The assignments shown in Figure 3, Figure 4 and Figure 5 above follow the “approximate description” approach, which identifies the dominant contribution to the normal mode. This approach avoids the more detailed but often cumbersome description in terms of potential energy distributions among all oscillators involved in each normal mode.

### 2.3. Intensity Calculations: Photons vs. Neutrons

An interesting issue concerning the description of the vibrational spectra from discrete calculations using small cluster models is revealed from the comparison of calculated versus experimental infrared and Raman spectra. Figure 6 and Figure 7 show that the DFT calculations with the monomer already provide a quite reasonable prediction of the infrared and Raman spectra of APTS. The assignments of the relevant vibrational modes agree with those derived from INS spectroscopy. 

However, the most relevant issue concerning these calculated spectra is that they are nearly insensitive to the size of the cluster used in the calculation. In other words, the intensity changes observed in the calculated spectra with increasing cluster size—i.e., from monomer to dimers and trimers—are generally small and do not allow a clear identification of the best model, as was observed for INS spectra. A similar situation arises for the infrared and Raman spectra of MAPS and APTES, as shown in the Appendix A. 

This observation, which places INS spectroscopy in a unique position to evaluate the role of non-covalent interactions, stems from the different origins of band intensities. In INS, the intensity of a vibrational mode is proportional to both the atomic scattering cross-sections of the moving nuclei and the amplitude of atomic motions, as described in Equation (1) above. While scattering cross-sections are well-defined physical constants, atomic motions are a straightforward result of vibrational calculations. Even at a moderate quality level, vibrational calculations predict good-quality atomic displacements for the normal modes. On the contrary, for the optical techniques, the evaluation of the intensity of a vibrational mode requires the calculation of transition matrix elements of the dipole moment and polarizability tensor operators for infrared and Raman, respectively, between the initial and final states [34,35]. This is computationally challenging, even considering the typical double harmonic approximation, in which the nuclear potential is assumed to be purely harmonic [34,35], and the relevant properties are expanded to the first order in the nuclear coordinates. For instance, since the polarizability itself is the second derivative of the energy with respect to an applied electric field, the Raman intensity is a third-derivative property [34]. 

The impact of the non-covalent interactions in the calculated infrared and Raman intensities is expected to decrease with decreasing strength of the interaction, i.e., from strong hydrogen bonds to dispersion van der Waals interactions. As a result, intensity calculations for the optical techniques can yield good-quality predictions when considering covalent bonding in molecular systems but provide a poor description of systems when multiple non-covalent bonding interactions must be considered.

## 3. Materials and Methods

Compounds: (3-Aminopropyl)trimethoxysilane, APTS (97%, IUAPC name: 3-trimethoxysilylpropan-1-amine, CAS 13822-56-5), [3-(Methylamino)propyl]trimethoxysilane, MAPTS (95%, IUPAC name: N-methyl-3-trimethoxysilylpropan-1-amine, CAS 3069-25-8) and APTES (99%, IUAPC name: 3-triethoxysilylpropan-1-amine, CAS 919-30-2) were purchased from Sigma-Aldrich (Gillingham, Dorset, UK) and used as received.

Vibrational spectroscopy: Inelastic neutron scattering experiments were performed with the TOSCA spectrometer [23,33,36], an indirect geometry time-of-flight spectrometer at the ISIS Neutron and Muon Source at the Rutherford Appleton Laboratory (Chilton, UK) [37]. The samples, with a total mass of ca. 2 g, were packed inside flat, thin-walled indium wire, sealed aluminum cans 5 cm in height by 4 cm in width, with a path length of 1 mm. These were mounted perpendicular to the beam using a regular TOSCA center-stick. Equation (1) contains an exponential term known as the Debye–Waller factor, which is partially determined by the thermal motion of the sample. By cooling the sample, this thermal motion can be reduced, leading to better results; for this reason, neutron spectra are usually obtained at temperatures below 20 K. In this work, the liquid samples were “shock-frozen” via quenching the filled aluminum can in liquid nitrogen before placement in the beam path. This procedure preserves the room-temperature morphology of the liquid, as shown previously [38]. Spectra were measured for the 16 to 8000 cm^−1^ energy-transfer range with resolution ΔE/E ≈ 1.25%. Data were converted to the conventional scattering law, S(Q,ω) vs. energy transfer (in cm^−1^), using the MANTID suite (version 6.6.0) [39].

Infrared spectra (50–4000 cm^−1^) of the liquids were measured at room temperature using a Bruker^®^ VERTEX 70v Fourier transform infrared spectrometer and a Bruker Diamond ATR accessory. All spectra were recorded at 4 cm^−1^ resolution with 256 scans and 8× zero filling to improve the peak shape. The spectra were corrected for the wavelength-dependent pathlength of ATR using the software package OPUS (version 8.7.10) from Bruker Optik GmBH (Ettlingen, Germany) (2020). 

FT-Raman spectra (50–3600 cm^−1^) of the liquids in quartz cells were recorded at room temperature using a Bruker MultiRam FT-Raman spectrometer with 500 mW laser power at 1064 nm and 4 cm^−1^ (16 scans) resolution was used with 8× zero filling. 

Quantum mechanical calculations: Density functional theory (DFT) calculations were carried out for single molecules and small clusters using Gaussian 09 software [40], using the built-in M06-2X functional with the 6-311+G(d,p) basis set. Geometry optimizations and frequency calculations were performed using the standard methods in Gaussian 09. Geometry optimizations were performed using the gradient method, and the final gradient length was less than 1 × 10^−4^ hartree bohr^−1^ or hartree rad^−1^, yielding geometries accurate to 0.05 pm or 0.1°. Frequency calculations were performed using analytical derivatives, within the harmonic approximation. All the optimized structures were found to be real minima, with no imaginary frequencies. For calculated Raman and infrared spectra, vibrational frequencies were scaled by a factor of 0.967 [41]. The Raman intensities were obtained from the calculated Raman activities, considering T = 298 K and ν_0_ = 9800 cm^−1^. The inelastic neutron scattering simulated intensities were estimated from the calculated eigenvectors using AbINS package [42], as implemented in the MANTID suite 6.6.0, and frequency values were not scaled. 

## 4. Conclusions

The results support the computational spectroscopy approach to interpret vibrational spectra of molecular systems. In the absence of crystalline structures—which enable calculations using periodic boundary conditions, i.e., simulating a perfect crystal—discrete calculations are the correct option to assess amorphous, or liquid-like, samples. Using a single molecule can lead to a reasonable description of the experimental spectra but ignores the role of non-covalent intermolecular interactions. In the systems herein discussed, the inclusion of intermolecular interactions using a cluster model is revealed to be fundamental for describing the INS spectrum of the compounds. Moreover, a trimer model, considering the possible hydrogen-bond contacts in the solid (such as N–H···N, N–H···O, and C–H···N and C–H···O hydrogen bond contacts), provides a remarkable description of the observed INS spectrum for the three samples. This validates the model, gives information concerning the relevant intermolecular contacts in the solid, and allows the accurate assignment of the vibrational spectra.

One relevant issue raised in this report is the lack of sensitivity of the calculated infrared and Raman spectra to the size of the cluster used in the calculation. The intensity changes observed in the calculated spectra with increasing cluster size (from monomers to dimers and trimers) are generally small. This does not allow for a clear identification of the best model, as was observed for the INS spectra. It stems from the difficulties in the calculations of intensities, which, for the optical techniques, are dependent on a demanding evaluation of the change in electronic properties during vibrations. This emphasizes the relevance of INS spectroscopy in the assessment of non-covalent interactions.

## Data Availability

Data are contained within the article and Appendix A.

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
