# Peer review of "Intermolecular Interactions in 3-Aminopropyltrimethoxysilane, N-Methyl-3-aminopropyltrimethoxysilane and 3-Aminopropyltriethoxysilane: Insights from Computational Spectroscopy"

_ijms, 2023, doi:10.3390/ijms242316634_

Round 1

Reviewer 1 Report

Comments and Suggestions for Authors

The manuscript is reporting on new scientific results obtained for three oxysilane compounds by INS spectroscopy and DFT calculations.  INS is good for the assessment of non-covalent interactions (mostly hydrogen bond contacts) but requires the use of advanced instrumentation operating at temperatures below 20 degrees kelvin. The experimental spectrum is in general comparable with the spectra calculated for the oxysilane monomer, dimer and trimer.  In this sense, the authors are commended for their computational skills.  Whereas Figure 2 is evidence for the complementarity of INS, FTIR and Raman, the two photon absorption/scattering methods can produce spectra of significantly higher signal-to-noise ratio than the neutron scattering method.  I guess this is where DFT calculations can be a better option than INS.

Technically, several improvements to the manuscript are needed:

1.  An emphasis of the scientific importance of studying non-covalent interactions in the Introduction with cited references.

2.  A better discussion of clusters larger than trimers.

3.  Statistical estimation of errors for the numerical results presented in Table 1, plus a clarification of HH and HT.

4.  An explicit indication (on p. 5) of vibrational modes that are readily observable with INS.

5.  An explanation of 'the atomic displacements in normal modes are a straightforward result of the computational approach to vibrational frequencies' (on p. 5).

6.  An elaboration on 'This discrepancy can be ascribed to the limitations of the cluster model, which allows the presence of “free” CH3 rotors' (on p. 6).

7.  More description of 'additional conformational freedom of the APTES skeleton' (on p. 7).

8.  Removal of (???) from the heading of section 2.3 (on p. 8).

9.  Addition of a couple references to support 'nuclear potential is assumed to be purely harmonic' (on p. 10).

10.  Deletion of the extra phrase 'at the STFC Rutherford Appleton Laboratory (Chilton, UK)' [43] (on p. 11).

11.  A suggestion how to deal with 'does not allow for a clear identification of the best model' (on p. 12).

Reviewer 2 Report

Comments and Suggestions for Authors

The Nolasco et al., have studied a computational spectroscopy approach for assignment of the inelastic neutron scattering spectra of the three title alkoxysilane derivatives. The authors performed simulations to match with the experimental spectra of entitled compounds. The authors spotlight the significance of incorporating non-covalent interactions, encompassing hydrogen bond contacts, to achieve a comprehensive understanding of the system. This manuscript has some serious concerns such as (i) comparison with other similar systems available in the literature, and (iii) interpretation of results need further convincing explanation/arguments.  However, this work may be recommended to publish after careful revision considering the below points and the revised version should be reviewed again before the publication.

1. In Figure 1, what is deciding factor for such kind of monomer arrangements? Did the authors consider other possibilities?

2. It is not sure how and what units of Y-axis were taken? I guess Gaussian 16 gives Raman Activities not intensities.

3. The Equation 1 is very superficial and does not provide exact solution to final intensity of band even given references are not helpful there. So, I recommend the authors to provide a working sheet about what parameters were taken as inputs and what were final output units.

4. As authors have focused on intermolecular interactions so in Figure 1, their intermolecular distances must be labeled with optimized values.

Comments on the Quality of English Language

No comments

Reviewer 3 Report

Comments and Suggestions for Authors

This manuscript shows experimental and simulation results about the vibrational spectrum of 3 organic compounds and contains inelastic neutron scattering, infrared and Raman data.

The approach to the subject is standard, but the results are relevant and one of the main conclusions of the paper (the relative insensitivity of optical spectroscopy to non-covalent interactions) is worthy of being highlighted. The paper is well written, the style is clear and the results are convincing, so I think that it deserves publication in your journal.

I do not have any major objection to the manuscript, but I would like to bring to the attention of the authors a few remarks that they could take into consideration and perhaps improve the paper:

1. It is not fully clear how many dimers and trimers were tested and how these are obtained. Is it just those shown in Fig. 1? Are there not other possible stable structures or have those been discarded? Would it not be interesting to compare the energies of different dimers and trimers for a given compound?

2. In my opinion, the most interesting finding is the inability of the monomer to reproduce correctly all the features observed in the INS spectra. This is also discussed in the test when describing the spectra for MAPTS and APTES. So I find disturbing that the corresponding calculated spectra for the monomer (and in fact also for the dimers) are not shown in Figs. 4 and 5.

3. The materials and methods section describes the INS and DFT techniques, but no details about the IR and Raman experiments is given.

4. I don't fully agree with the last paragraph of section 2: "The impact of the non-covalent interactions in the calculated infrared and Raman intensities is expected to decrease with decreasing strength of the interaction, i.e., from strong hydrogen bonds to dispersion van der Waals interactions. As a result,  intensity calculations for the optical techniques can yield good quality predictions when considering covalent bonding in molecular systems but provide a poor description of systems when multiple non-covalent bonding interactions must be considered.".

I agree with the remarks of the authors pointing to the additional complexity from the theoretical point of view when calculating optical spectra compared to the calculation of INS spectra. As they say, this is computationally challenging, but there have been enormous progress in this respect in the last years. However, from my point of view the problem is not that the calculations are failing, but rather that the result obtained with the simplest approximation (the monomer) reproduces already quite well the experimental data, making those insensitive to finer details. Probably, in order to progress, one should develop a reasonable way of quantifying the agreement between experiment and calculation, but I am not aware of
any "figure of merit" proposed in the field.

Below I add a list of typos or suggestions that I noted while reading the manuscript:

1. In Scheme 1, the last letter of (3-aminopropyl)triethoxysilane is cut.

2. In page 2, join references [7-16][17-21] into [7-21] and close parenthesis.

3. P. 2: Its application has a thermal insulator --> Its application AS a thermal insulator

4. P. 2: notable improvements in spec-trometer sensitivity– e.g., TOSCA at ISIS [22] and VISION at Oak Ridge [23] –
   I think that the Lagrange instrument at ILL could (should?) be added here.

5. P. 2: Add comma between APTS and MAPTS in the last paragraph.

6. Caption of table 1: the shortest is show --> the shortest is shown

7. Eq. (1) appears badly placed

8. P. 5: i should appear as subscript in sigma_i and U_i.

9. Caption of Fig. 4: same heat-to-head trimer --> same HEAD-to-head trimer, and cm^-1 cut in line.

10. P. 8: Remove question marks in heading 2.3

Comments on the Quality of English Language

English is good. Just a few types to correct (see above).

Round 2

Reviewer 2 Report

Comments and Suggestions for Authors

Revision is acceptable. I still insist the authors  that it would be more appropriate if authors label distances in Figure 1.

Comments on the Quality of English Language

No